# Design and Field Test of a Mobile Augmented Reality Human–Machine Interface for Virtual Stops in Shared Automated Mobility On-Demand

Fabian Hub [1,*] and Michael Oehl [2]

1   German Aerospace Center (DLR), Rutherfordstraße 2, 12489 Berlin, Germany
2   German Aerospace Center (DLR), Lilienthalplatz 7, 38108 Braunschweig, Germany
*   Correspondence: fabian.hub@dlr.de

**Abstract:** Shared automated mobility on-demand (SAMOD) is considered as a promising mobility solution in the future. Users book trips on-demand via a smartphone, and service algorithms set up virtual stops (vStop) where users then need to walk to board the automated shuttle. Navigation and identification of the virtual pickup location, which has no references in the real world, can be challenging. Providing users with an intuitive information system in that situation is essential to achieve high user acceptance of new automated mobility services. Our novel vStop human–machine interface (HMI) prototype for mobile augmented reality (AR) supports users with information in reference to the street environment. This work firstly presented the results of an online interview study (N = 21) to conceptualize an HMI. Secondly, the HMI was prototyped by means of AR and evaluated (N = 45) regarding user experience (UX), workload, and acceptance. The results show that the AR prototype provided high rates of UX especially in terms of high pragmatic quality. Furthermore, cognitive workload when using the HMI was low, and acceptance ratings were high. The results show the positive perception of AR for navigation tasks in general and the highly assistive character of the vStop prototype in particular. In the future, SAMOD services can provide customers with vStop HMIs to foster user acceptance and smooth operation of their service.

**Keywords:** virtual stop; shared automated mobility on-demand; HMI; augmented reality; field test; user-centered design

## 1. Introduction

In the quest for new forms of impactful and sustainable mobility solutions, automated vehicles (SAE levels 4 and 5) are likely to play a large role in future scenarios, especially in deployment with mobility-as-a-service algorithms (i.e., ride pooling) [1–3]. The advantages of such shared automated mobility on-demand (SAMOD) services are sustainability in terms of resource consumption as well as overall efficiency, accessibility, and flexibility from a user and traffic standpoint [4–8]. It is projected that SAMOD can contribute to more eco- and user-friendly modes of individual transportation in rural and urban areas, particularly in combination with public transportation and with sufficient support of local policies [9–11]. In contrast to conventional ride hailing, ride-pooling algorithms bundle trips of customers who want to go in the same direction [12,13]. This aspect results in maximized capacity and reduced miles traveled per shared automated vehicle (SAV) [14,15]. Consequently, this also means that customers are usually not picked up at their current GPS location or address. Instead, specific pickup locations are set up on-demand, mostly at large streets or intersections nearby, where the shuttle and user meet. This flexible stop is only digitally created as a kind of virtual infrastructure and is the meeting point for the customer and SAV for this particular ride [16,17]. The mobility operator's backend algorithms coordinate booking and availability of rides as well as the instantiation and activation of pickup locations across the citywide digital grid of virtual stops (vStop).

From an operator's perspective, various rules apply for setting up these digital facilities, e.g., the density of the digital vStop grid, acceptable detours, vehicle capacity, or service hours. Although vStops are digital, the local physical environment itself is taken into consideration, too, when managing the vStop grid [18,19]. With this approach, the SAV can remain on well-known major routes while customers bridge the waiting time by walking a short distance to that assigned pickup location. Currently, conventional ride-pooling operators such as Moia or ViaVan already use flexible stops. Customers are guided to the pickup location via digital mapping methods (such as Google Maps) displayed on their smartphones.

The main challenge at this stage of the user journey is to effectively navigate and identify the pickup location in a short amount of time without physical cues in the environment [20]. With solutions currently practiced by on-demand mobility providers, users seem prone to confusion and misinterpretation of information. Hence, conventional digital maps may not be the ideal way of information representation when users are trying to find a very specific position in the street environment. Circumstances such as unfamiliar territory, confusing traffic situations, or time pressure can lead to stress, mistakes during navigation, and delays of the users. Customers not finding the pickup location in time or even missing their ride can pose the risk of degrading the operation performance and negatively influencing the perceived reliability, punctuality, and overall user experience (UX) of the SAMOD services [21], not to mention leaving other passengers unnecessarily waiting in the SAV.

Furthermore, the lack of a human driver on board of the automated pickup vehicle leads to high requirements of information presentation on the user side when identifying the SAV. As of yet, the driver would recognize the waiting customer curbside and could accordingly react. In future scaled-up SAV deployment, user assistance is crucial in pickup situations because similar-looking SAVs (branding, vehicle type, color, etc.) may approach the same pickup spot. Although license plates are distinguishing marks, they are quite small and not very easy to recognize by customers in dense traffic, when the vehicle is in motion, or in foreign countries (e.g., different formats are used; vehicles in the US usually have only one license plate on the back, while vehicles in Switzerland have a very small license plate on the front).

Hence, it becomes essential to provide customers with significant information and high UX along the whole user journey in order to achieve high user acceptance of SAMOD. Otherwise, users are likely to not use automated mobility service. When targeting the challenges of navigating to the flexible pickup location and identifying the SAV, information representation must be clear and efficient and needs to be accessible to users. By providing an intuitive HMI that accompanies the user journey with meaningful information, the overall UX of the novel mobility services, and eventually user acceptance, can be increased [22].

### 1.1. The Concept of Virtual Stops (vStops) and Accompanying Human–Machine Interface

As mentioned before, the goal of increasing user acceptance and widespread adoption of SAMOD services requires profound information presentation in order to provide high UX in the first place.

Therefore, the concept of a vStop ought to be more than just an ad hoc meeting point for the SAV and user. Accordingly, vStops have two very important aspects: firstly, vStops are a user-centered human–machine interface (HMI) to clearly and efficiently provide users of SAMOD with information along the user journey so that they experience high levels of UX and safety while also reducing the levels of uncertainty and discomfort [23]. Secondly, vStops act as a sort of virtual traffic coordinating entity that manages the local traffic space during a pickup situation with the help of intelligent traffic infrastructure. The behavior of various traffic participants in the local traffic space can be adjusted by vStops through locally enforcing rules at interplay with other traffic infrastructure elements. With this functionality, vStops will enable efficient pickup scenarios in the future [17].

However, this paper focused on the perspective of vStops as a mobile HMI. From a user-centered point of view, UX with SAMOD services is largely driven by information accessibility along the user journey, in particular in the scenarios prior to boarding the SAV. By designing an HMI that intuitively provides relevant information in real time to the user, uncertainty during navigating the pickup location can be reduced, high levels of UX can be achieved, and the user feels in control of his or her trip. Furthermore, users not having the right information intuitively presented can compromise a smooth process of ride pooling. The effectiveness of SAMOD is highly dependent on users being positioned at the pickup location in time to board the vehicle right away. These aspects highlight the need for a user-centered vStop HMI of high pragmatic quality (as a subdimension of UX) along the user journey [24]. Hence, the impact of new sustainable automated driving modes is driven by their UX and intuitive information supply, especially in the pickup scenario.

### 1.2. Augmented Reality and Pedestrian Navigation

This work centered on the means of augmented reality (AR) because of its characteristic of enhancing the physical environment with digital content. This can be beneficial to users when navigating to the pickup location as well as identifying the vStop and approaching SAV.

A central definition for AR was given by Azuma, who described it as a technology with three key characteristics: (1) it combines real and virtual content; (2) it is interactive in real time; and (3) it is registered in three dimensions [25]. AR technology is currently used in various domains such as marketing, medicine, education, and entertainment [26].

Although AR is not restricted to visual stimuli only, this work focused on visual AR modalities [27]. The visual modalities of AR technology can be experienced with head-mounted displays, smartphones, tablet computers, or desktop computers. Virtual images and objects are registered and displayed over local, physical objects in order to enrich the environment with information using the methods of computer graphics. Aiming at easy accessibility and a high likelihood of rapid implementation, this research utilized AR for handheld mobile devices (mobile AR), such as smartphones. The advantages of mobile AR systems mainly lay in ergonomics and can be summarized as being lightweight and comfortable and natural to use, and hence little fatiguing for the users [28]. Smartphones are close to the context of use in this study. Among other mobile devices, smartphones also show the most constant AR performance in terms of immersion and usability [29]. From a user perspective, the interaction with mobile AR is determined by two major aspects. Comprehensibility, on the one hand, is driven by the perception capabilities of the system (e.g., tracking latency, illegibility due to ambient light, underestimated/overestimated depth). On the other hand, the system's manipulability describes the ergonomics factors (e.g., weight, bulkiness, and feedback) and how users interact with the device [30]. Both dimensions of mobile AR interaction need to be at an adequate level to provide comfortable spatial interaction to users.

In directly presenting content in reference to the environment, AR can be beneficial for users of SAMOD in overcoming the challenge of getting to the pickup location in time. Accessing and understanding essential information such as a pathway and pickup location can easily improve a user's overall experience, competence, and performance when getting to flexible vStops. Research of usability has shown that consuming information requires low rates of cognitive workload when it is presented by means of AR in the appropriate context [31]. In contrast to conventional digital map-based solutions, AR leaves the brain capacity for secondary tasks because it is very easy to navigate with [32]. This can be beneficial in real-world conditions and especially during a navigation task in roadside environments, which is the case for vStops.

In general, users hold their device upright, and the augmentation of the real world is achieved by superimposing digital elements that are helpful for navigation (e.g., a digital path or pointing arrows) on a live-stream video (video see-through). Users can freely move and do not need further equipment than the smartphone (equipped with an orientation

sensor, GPS sensor, and camera). A first AR concept for pedestrian navigation was studied in 2005 by Narzt et al. [33]. In addition, alternative implementations to the video see-through approach have been conceptualized and researched [34–36]. However, different forms for pedestrian navigation, such as landmark-based concepts, are not considered further because this work focused on the video see-though approach of AR.

The use case of AR for pedestrian navigation application has also been a topic to research lately and showed potential to improve UX in navigation tasks [37]. AR possesses the potential to improve the experience and performance of users during the navigation task by increasing attentiveness and spatial awareness during task solving [38]. The question of high UX of AR navigation for the present use case in this study of getting to a vStop is yet unanswered.

In a nutshell, recent studies have shown the advantages and disadvantages of AR technology for users' navigation purposes. AR applications (not only) for navigation are, to a great extent, dependent on technological advances in the field of AR [39]. Accordingly, the latest outcomes depend on factors such as the available technology readiness level and methodological approaches. Rehrl et al. (2012) conducted an in situ field study to test the UX and performance of a self-implemented, location-based AR pedestrian navigation against other modes of navigation, such as a digital map and voice. The results showed that the AR system was characterized by lower rates of UX and performance by the participants compared with voice or digital map navigation [40]. Another comparative study of UX and performance between digital map navigation and a self-developed AR prototype was conducted by Brata and Liang [41]. They found that digital maps provided higher pragmatic quality to users in a field study. Nevertheless, users valued the AR prototype for navigation in terms of hedonic quality and overall performance. Another study compared a standardized, commercially available AR application (Baidu) for pedestrian navigation against a digital map from the same service provider in terms of performance and gazing. No significant performance and gazing differences were found, but the results indicate that AR can enhance the attention paid to surrounding traffic participants as the salience of moving objects was increased [42].

Besides available technology and different navigation tasks (length, complexity, circumstance, etc.), one reason for the diverging results also lays in the lack of standardized AR interface design for navigation purposes. Most of the investigated AR solutions were self-developments by the researchers. The design processes were mostly nontransparent, except for one study found [41,43]. Therefore, it is very important to present the subsequent evaluation of an AR interface for pedestrian navigation in reference to the forgoing design process. This paper addressed this research gap by presenting HMI development and testing in one joint work, consisting of two user studies.

### 1.3. Research Question

In future SAMOD scenarios, customers will need efficient information supply via mobile devices to seamlessly reach their vStop and with high rates of UX. In this use case, AR can be seen as the appropriate technology to present valuable information in reference to the environment and therefore to assist users along the user journey. Hence, this paper's goal was to present a user-centered vStop HMI concept that effectively increases competence and reduces uncertainty for users when navigating to vStops.

With methods of user-centered design, this paper tried to answer the following research question: How should a vStop HMI be designed from a user-centered perspective so that user experience (UX) during the pickup scenario of SAMOD is maximized?

This work was composed of two user consecutive studies of explorative character that follow the same research question.

Firstly, conceptualizing an efficient augmented reality (AR) interface with the most valuable information elements was aimed. In order to do so, a first user study was conducted as an online interview study. Generalized AR information entities for this present

vStop use case were presented and evaluated by participants regarding comprehensibility, importance, usability, and preference.

Secondly, a first mobile vStop prototype was created by means of AR based on the results of the first interview study containing the most supportive AR information elements. In a field test, the vStop prototype was evaluated regarding overall UX, workload, and acceptance. In addition, qualitative user feedback was captured.

With this approach, the process of developing the HMI and its performance were investigated. Consequently, joint conclusions were drawn on the prototype's design. Finally, core usability issues and further for vStop HMI design requirements were identified.

## 2. First User Study: Identifying Most Important AR Information Elements

The following section thoroughly describes the first user study. AR can simplify how users get to their pickup location and identify the approaching vehicle. Hence, the goal of the first interview study was to identify the most relevant information elements of or combinations for a vStop HMI for an effective mobile AR interface. The study focused on the user journey scenarios of navigating to the pickup location, identifying the vStop, and identifying the approaching SAV. The foundations for this user study were the results from an earlier conducted expert workshop that identified the necessary information entities [44].

### 2.1. Method

This subsection elaborates on the online interview study participants, used variables, study design, procedure, and used stimulus material.

#### 2.1.1. Participants

Interviewees were recruited over university channels, e.g., faculty subject pools and social media (e.g., Facebook groups). A group (N = 21, women = 14) of relatively young adults (age M = 23.62; SD = 2.77) participated in the online interview study. All but one participant were students. All participants were German and lived in large cities (more than 200.000 inhabitants). Ten subjects lived in a metropolitan area (more than 3 million inhabitants). Ninety percent of the interviewees were in possession of a driver's license. All but one participant were familiar with the topic of automated vehicles, and the overall interest in automated vehicles was about average (M = 3.42; SD = 0.90, on a 5-point Likert scale, 1 = "not at all" to 5 = "very strong"). Conventional on-demand ride pooling was used in private circumstances by 5 participants before. The services used were Berlkönig (n = 3), Moia (n = 1), and CleverShuttle (n = 1), which operate in large cities in Germany.

The majority (71%) was familiar with virtual reality (VR), and 38% experienced AR before. Only 3 participants had never experienced any mixed reality information technology (AR or VR). Participants' mean technology affinity was slightly above average (ATI-score = 3.70; SD = 0.89, on a 6-point Likert scale, 1 = "do not agree" to 6 = "fully agree") [45]. The study was conceptualized and realized in accordance with the Declaration of Helsinki. Informed consent was obtained from all participants before the experiment. The participants were allowed to stop the study at any point without justification or consequence. The participants volunteered but were financially compensated with EUR 10.

#### 2.1.2. Independent Variables

The user journey scenarios of interest were navigating to the pickup location, identifying the vStop, and identifying the approaching SAV. For each scenario, different potential interfaces with different information elements (as for information strategies) were shown.

In the first scenario, the information elements were as follows: First, an *orientation* element (i.e., a compass-like function to show the walking direction) was shown, visualized as an arrow object pointing at the direction to walk. Second, a *path* element (i.e., directly showing the way to walk) was shown as a line on the ground. Third, a *turning* element (i.e., showing the node where and in which direction to turn next) was presented as a distantly floating double-arrow object.

The second scenario showed 4 different information elements. First, it had the same *orientation* element (i.e., showing the direction to the pickup location) as in scenario 1. Second, an area on the *floor* (i.e., showing the location by highlighting the ground) was presented. Third, a *floating* symbol (i.e., showing the pickup location by placing a flying element above it) and, fourth, a *standing subject* (i.e., to identify the pickup location with a digital 3D model) were shown in scenario 2.

The 4 AR information elements to identify and board the right SAV (scenario 3) were as follows: First, an *object highlighter* (i.e., an element to highlight the approaching vehicle); second, a vehicle *stopping area* (i.e., the position where the vehicle will stop is highlighted on the street); and third, a customer *waiting area* (i.e., the position where the user can wait and directly board the SAV is highlighted on the sidewalk) were shown. Additionally, the *orientation* element (i.e., showing the direction where the SAV is approaching from) was also included as the fourth information element.

### 2.1.3. Dependent Variables

The single information elements were evaluated regarding *comprehensibility* (on a 7-point Likert scale; 1 = "not comprehensible at all" to 7 = "very comprehensible") and *importance* (on a 7-point Likert scale; 1 = "not important at all"; 7 = "very important") to solving the task. Both items were chosen in the first interview phase of each scenario to make sure that the information strategies themselves were helpful and understood by the participants.

In the second interview phase, possible interface solutions (i.e., combinations of AR information elements) were rated according to a self-constructed usability score, consisting of the standardized UEQ usability subscale items *efficiency* and *comprehensibility*. A 7-point Likert scale was used to comply with UEQ standards [24].

Lastly, users' favorite interfaces were identified by forced choice of the most desirable AR interface in familiar and unfamiliar territory.

To determine the AR HMI that effectively supports the user and reduces uncertainty when getting to vStops, the following criteria were set by the researchers: An AR interface (a combination of information elements) had to pass the threshold of an empirical mean value of > 6.00 in the self-constructed usability rating. Additionally, it also needed to be ranked among the top three in participant-voting for the most desired AR information in familiar and unfamiliar territory. Additionally, the coherence of an HMI for all three scenarios was taken into account.

### 2.1.4. Materials

Low-fidelity stimulus material was prepared in the form of static pictures that showed smartphone mock-up screens to visualize a potential AR application.

In general, 3 scenarios of the user journey were visualized as smartphone mock-up. Every smartphone mock-up screen was identically structured in each scenario and split into two parts. In the bottom part (the non-AR part), conventional service information of on-demand ride-pooling services was placed, such as time to pick up and remaining distance to and address of the vStop. This approach was chosen to give a realistic impression of a mobility provider app in the three scenarios. The upper part of the screen was dedicated to the AR functionality. In each scenario, the upper part of the smartphone screen would show a typical camera image of the respective situation, as if one would look through the smartphone (the video see-through AR approach). A walking street (first scenario), curbside pedestrian zone (second scenario), and vehicle approaching curbside (third scenario) were shown as backgrounds. The backgrounds were held in black and white to not distract the participants from the presented AR elements. The elements were placed in the smartphone field of view as overlay. The elements were of light blue color to be easily recognizable even for users with conditions of colorblindness (see Figure 1).

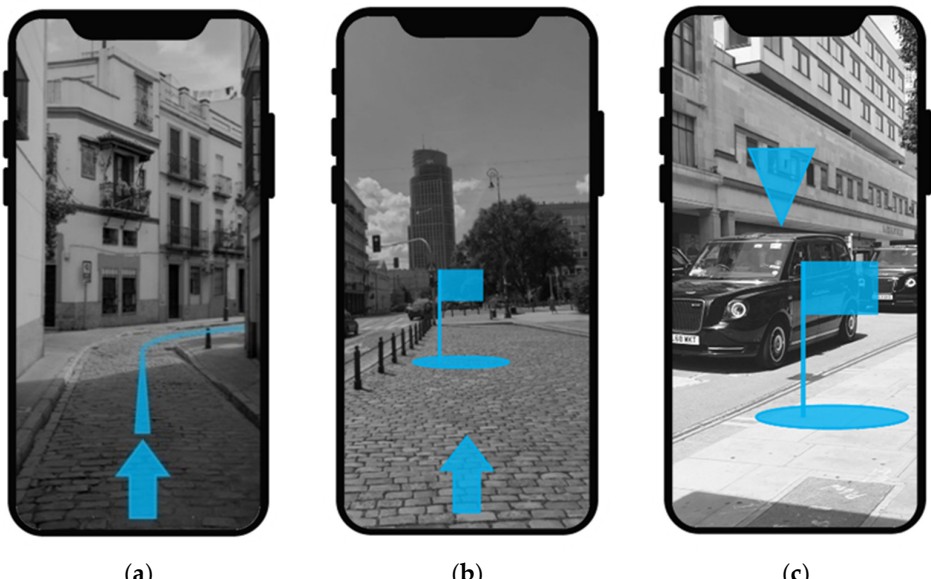

**Figure 1.** AR interface for the 3 phases of the user journey: (**a**) navigation to the vStop with orientation (added post hoc) and pathway element; (**b**) identifying the vStop with orientation and standing subject element (waiting area was included post hoc); and (**c**) identifying SAV with waiting area and object highlighting element (standing subject element was included post hoc).

The presented elements stood for different information strategies that were ideated and visualized during the previously conducted expert workshop [44].

### 2.1.5. Study Design

The user study was conceptualized as an online interview study with a standardized questionnaire and quantifiable items. The study was conducted via a web-conferencing service (Skype for Business). For documentation reasons, all interviews were recorded. The participants received all the necessary documentation (e.g., declaration of consent and data protection declaration) upfront per mail. The stimulus material was presented to the participants via a collaborative online tool (miro.com). The participants were asked to share their screen throughout the interview so that the interviewer was able to control what the participants were looking at. The participants' answers were directly recorded into an MS Excel sheet by the interviewer. The average duration of the interviews was about 45 min.

### 2.1.6. Procedure

The interview was structured as follows: First, the participants answered questions regarding demographics. Then, the research topic and objectives were presented. A short introduction into AR was given using an example of an existing AR application. The users where asked to put themselves into the fictive situation of using a SAMOD service and having already booked a SAV via a smartphone. They would now need to navigate to their vStop location. The task was divided into 3 consecutive scenarios. In the fictive situation, the users would have a smartphone with AR capabilities at hand, and by the aid of the AR interface, they would get to the vStop to board the vehicle. The elements of the smartphone screen mock-ups were thoroughly described, and the objective of this interview was clarified. The participants were told not to focus too much on the actual look of the information elements (object shapes, etc.) but rather the informational character behind it. Accordingly, potential variations of some elements were also shown to underline that all provided visualizations were quick sketches and focused on visualizing a distinctive information strategy.

The investigated situation comprised 3 scenarios of interest along the SAMOD user journey. The first scenario was the user's navigation to the pickup location. The user

standing on the side of the street and wanting to find and identify the exact position to board the vehicle was the second scenario. In the third scenario the booked SAV approached the vStop, and the user would need to identify the vehicle. Each of the 3 scenarios was structured into 3 parts of questioning:

(1) Evaluating fundamental single information strategies of an HMI. All elements were individually shown, explained in detail, and evaluated by the participants. The smartphone mock-up screens only showed the blue information elements and the grayscale background (service information was covered to avoid distraction). With this approach, the information strategy itself was rated.

(2) Usability ratings of all possible information element arrangements, as potentially finished AR interfaces, were investigated. Accordingly, combinations of different information strategies were systematically presented. The information presentation of the mock-ups ranged from simple to complex, starting by only showing one element and ending with a combination of all possible information elements, showing all information strategies combined in one interface. Accordingly, the first scenario contained 7 different interfaces, scenario 2 contained 15 different interfaces, and scenario 3 contained 15 different interface possibilities.

(3) The interview ended with the choice of a preferred HMI when being in familiar and unfamiliar territory. All potential interfaces of possible AR-element combinations were simultaneously presented to the participants to choose a favored interface for two different hypothetical circumstances: being in a foreign city and being in a familiar environment. Differentiation between these two circumstances was chosen to gain deeper understanding of alternating information requirements by users.

### 2.2. Results

In the following, the results of the first interview study are presented. Firstly, comprehensibility and importance ratings of the AR information strategies are shown for each of the three investigated scenarios. Secondly, this subsection reveals the usability ratings of the different interfaces, consisting of systematic arrangements of AR elements. This subsection ends with the user preferences for an HMI, meaning the most desirable combination of information elements on a mobile AR interface for two hypothetical situations (in familiar and unfamiliar territory).

### 2.2.1. Data Analysis

Interview data and the results of evaluating the AR information element by the participants were descriptively analyzed. No inferential statistic tests were conducted due to the small sample size. Mean values, standard deviation, as well as the highest and lowest values of the *comprehensibility* and *importance* ratings were calculated for all shown AR information elements. For identifying the interface with the highest usability, all *usability scores* of the possible information element combinations were calculated for each of the three investigated scenarios. Eventually, all information element arrangements that received a vote for favored AR interface were counted with respect to familiar and unfamiliar territory.

### 2.2.2. Comprehensibility and Importance of AR Information Elements

The first scenario was about a user navigating via a mobile HMI to the flexible pickup location where he or she will be picked up by the SAV.

The single AR information that received the best evaluation in comprehensibility (M = 6.81; SD = 0.39) and importance (M = 6.29; SD = 1.03) was the *path* element that focuses on highlighting the route. In the second scenario, users needed to find the exact pickup location curbside, where they can wait and get ready to board the SAV. The *standing subject* (represented by a flag) was evaluated best in comprehensibility (M = 6.90; SD = 0.29) and importance (M = 6.48; SD = 0.73). The *object highlighter* (represented with a floating triangle above the vehicle) was the most comprehensible (M = 6.33; SD = 1.13) and most important

(M = 6.33; SD = 0.71) single information element when identifying the approaching SAV (scenario 3). All results are shown in Table 1.

**Table 1.** Comprehensibility and importance rating of AR information elements by participants of the first user study (N = 21).

| AR Information Element | Comprehensibility M (SD) | Importance M (SD) |
|---|---|---|
| *Scenario 1: Navigation task (route)* | | |
| Path | 6.81 (0.39) | 6.29 (1.03) |
| Orientation | 5.57 (1.40) | 6.05 (1.05) |
| Turning node | 5.14 (1.36) | 5.48 (1.05) |
| *Scenario 2: Identification task (place)* | | |
| Standing subject | 6.90 (0.29) | 6.48 (0.73) |
| Floor | 6.43 (0.85) | 6.05 (1.21) |
| Orientation | 4.52 (1.43) | 4.90 (1.54) |
| Floating | 3.81 (1.62) | 4.62 (1.70) |
| *Scenario 3: Identification task (vehicle)* | | |
| Object highlighter | 6.33 (1.13) | 6.33 (0.71) |
| Stopping area | 5.10 (1.23) | 4.52 (1.62) |
| Waiting area | 4.86 (1.61) | 4.14 (1.67) |
| Orientation | 4.00 (1.51) | 3.95 (1.59) |

### 2.2.3. Usability of AR Interfaces

In the first scenario, two possible AR interfaces reached the threshold of a mean value of 6.00 in usability rating (see Table 2). A combination of arrow overlay for *orientation* and the *pathway* information was evaluated with a score of 6.29 (SD = 0.86). The interface with the highest score (M = 6.79; SD = 0.37) showed the *path* as single information. A combination of all elements received the lowest mean value evaluation.

Four different AR interfaces reached the threshold for the usability score in the second scenario. The interface solely showing the *standing subject* (M = 6.64; SD = 0.42) was evaluated highest regarding mean usability. The interfaces that showed only the *floor* (M = 6.26; SD = 0.75), a combination of *orientation arrow and standing subject* (M = 6.21; SD = 1.08), and a combination of *orientation arrow and floor* (M = 6.17; SD = 0.94) were also evaluated with high mean usability scores. The interface that was evaluated with the lowest mean usability showed the *floating* indicator for the pickup locations. In addition, none of the interfaces containing the *floating* element passed the threshold of 6.00 in mean usability. The participants stated that they were not able to really tell the exact location with this information element.

In the scenario to identify the approaching SAV (scenario 3), the critical score of 6.00 was reached by two potential AR interfaces for a vStop HMI. The interface only directly highlighting the *object* (M = 6.36; SD = 0.87) was evaluated with the highest mean usability in that scenario. Adding the *waiting area* highlighter to the *object* highlighter (M = 6.02; SD = 0.91) also reached very high mean usability ratings. As in the first scenario, the interface with all potential AR information elements received the lowest usability score.

### 2.2.4. Users' Favored AR Interfaces

The most favored combination of AR information elements to navigate to the pickup location (scenario 1; seven interfaces) was identical in familiar and unfamiliar territory but with a different ranking. In unfamiliar territory, the combination of *path and turning node* (38%; 8 of 21 participants) was the most favored information element arrangement, followed by *path and orientation* (29%; 6 of 21 participants) and *pathway* information only (24%; 5 of 21 participants). However, in familiar territory, the single *path* information element (42%)

was chosen as the favored interface by the participants. *Path and orientation arrow* (21%) and *path and turning node* (16%) followed.

**Table 2.** Usability rating of arranged AR interfaces by participants of the first user study (N = 21).

| Scenario 1: Navigation Task (Route) | | Scenario 2: Identification Task (Place) | | Scenario 3: Identification Task (Vehicle) | |
|---|---|---|---|---|---|
| Interface Elements | Usability M (SD) | Interface Elements | Usability M (SD) | Interface Elements | Usability M (SD) |
| 1 AR information element | | | | | |
| Path | 6.79 (0.37) | Standing | 6.64 (0.42) | Object | 6.36 (0.87) |
| Orientation | 5.19 (1.04) | Floor | 6.26 (0.75) | Stopping | 4.98 (1.17) |
| Turning | 4.93 (1.12) | Orientation | 3.83 (1.15) | Waiting | 4.48 (1.26) |
| | | Floating | 3.81 (1.33) | Orientation | 4.00 (1.21) |
| 2 AR information elements combined | | | | | |
| Orientation + Path | 6.29 (0.86) | Orientation + Standing | 6.21 (1.08) | Object + Waiting | 6.02 (0.91) |
| Path + Turning | 5.90 (1.14) | Orientation + Floor | 6.17 (0.94) | Object + Stopping | 5.62 (1.30) |
| Orientation + Turning | 5.10 (1.55) | Floor + Standing | 5.67 (0.91) | Orientation + Object | 5.10 (1.20) |
| | | Floor + Floating | 5.26 (0.97) | Stopping + Waiting | 4.33 (1.69) |
| | | Floating + Standing | 4.81 (1.08) | Orientation + Waiting | 3.93 (1.38) |
| | | Orientation + Floating | 3.90 (1.37) | Orientation + Stopping | 3.81 (1.26) |
| 3 AR information elements combined | | | | | |
| Orientation + Path + Turning | 4.48 (1.50) | Orientation + Floor + Standing | 5.81 (0.94) | Object + Stopping + Waiting | 4.79 (1.16) |
| | | Orientation + Floor + Floating | 5.07 (1.49) | Orientation + Object + Waiting | 4.40 (1.44) |
| | | Orientation + Floating + Standing | 4.74 (1.02) | Orientation + Object + Stopping | 4.40 (1.52) |
| | | Floor + Floating + Standing | 4.55 (0.91) | Orientation + Waiting + Stopping | 3.52 (1.36) |
| 4 AR information elements combined | | | | | |
| | | Orientation + Floor + Floating + Standing | 4.19 (1.54) | Orientation + Object + Stopping + Waiting | 3.17 (1.54) |

In scenario 2 (identification of pickup location; 15 interfaces), the most favored AR interfaces in unfamiliar territory were *orientation arrow and standing subject* (29%), followed by *orientation arrow and floor* (24%) and a combination of *orientation, floor, and standing subject* (19%). In an environment that is familiar to the participants, the favored AR interface was also *orientation arrow and standing subject* (24%). The combination of the *orientation, floor, and standing subject* as well as the *orientation arrow* only (both 19%, 4 of 21 participants each) were also favorable.

The most desired combination of AR elements for a mobile vStop HMI for vehicle identification (among 15 interfaces) in unfamiliar territory was *object highlighter and waiting area* (48%), followed by *object highlighter and vehicle parking spot* (24%). *Object highlighter and waiting area* (43%; 10 of 21 participants) was also the participants' favorite in familiar territory, followed by *object highlighter and vehicle parking spot* (29%) and a single *object highlighting element* (19%).

It is worth mentioning that in none of the investigated scenarios, the maximum amount of information elements was preferred by the participants. However, still, on average, the participants slightly favored more information elements in an AR interface when being in unfamiliar territory.

2.2.5. AR Interface Selection for Each Scenario

AR interface selection was based on the overall best information element evaluation across all investigated aspects.

For the task of navigating to the pickup location, the interface with the highest mean usability score (M = 6.79; SD = 0.37) showed only an augmented *pathway* element. This potential interface received 24% of the favored vote in unfamiliar territory and was most favored (42%) in familiar territory. All other possible interfaces were evaluated less positive across all investigated aspects. For the second task (identifying the waiting area in an urban environment), the augmentation of a *standing subject* with an overlaying *orientation arrow* was desired most by the participants (29% in unfamiliar and 24% in familiar territory) and scored the highest in mean usability (M = 6.21; SD = 1.08). In order to have a consistent interface that spans across the first two scenarios, we decided to add the *overlaying arrow* to the navigation scenario interface. This particular combination of information elements was also among the most favored interfaces (29% in unfamiliar and 21% in familiar territory) and received a very high mean usability rating (M = 6.29; SD = 0.86), also exceeding the threshold of 6.00.

The third scenario was about identifying the approaching SAV in the street environment. The interface that combines the *waiting area* information and a floating *object highlighter* was the most favored one (48% in unfamiliar and 43% in familiar territory). This combination of AR information elements also scored very high in mean usability (M = 6.02; SD = 0.91), so that this interface design was chosen. Again, to smoothen the interface designs between scenarios 2 and 3, the augmented *waiting area* element was added to the interface in the second scenario. Although the combination of *overlaying arrow, augmented waiting area*, and *standing subject* (M = 5.81; SD = 0.94) did not reach the 6.00 threshold, it was among the top three interface designs for scenario 2 (19% in unfamiliar and unfamiliar territory). The popular choice of the participants gave justification for that decision. In addition, the *standing subject* was added to the third scenario post hoc to the interface of identifying the vehicle in order to be consistent with the AR information elements of scenario 2 (see Figure 1).

## 3. Second User Study: Evaluating the vStop HMI in a Field Test

The following section describes the second user study as a field test in detail. The goal of this study was to evaluate the vStop AR prototype in unfamiliar territory regarding overall usability, acceptance, and workload during task completion, meaning the navigation to a fictive vStop. Additionally, qualitative user feedback was captured to understand the handling and usability of the HMI in real-life exposure. The HMI prototype was developed based on the results of the first user study. By this approach, the gained insights from the static stimulus material of the online interview study were put to test in a live setting with a functional prototype.

### 3.1. Method

This subsection goes into detail about the user study participants, used variables, study design, procedure, and developed vStop HMI prototype.

### 3.1.1. Participants

A total of 30 men and 15 women participated in the user study to evaluate the vStop AR prototype in a field setting. Eighty-six percent were unfamiliar with the test area, and only 2 participants knew the location. The group was relatively diverse in terms of age (ranging from 19 to 58 years), and the average age was 33.87 years (SD = 11.65). Sixty-nine percent lived in large cities (more than 100.000 inhabitants). The participants were mainly students (40%) or employed (42%). Two participants were already retired. Almost half of the participants had a university degree (49%). All participants but one were in possession of a driving license. Sixty-one percent of the participants stated that they drive less than 10.000 km per year. All participants were familiar with automated vehicles (SAE levels 4 and 5), and 4 participants had already experienced a ride in an AV. The overall interest in AVs was very high (M = 4.13; SD = 0.88, 5-point Likert scale, 1 = "not at all" to 5 = "very strong"). Only 4% of the study participants experienced conventional on-demand

ride pooling before and stated a very infrequent use (once: n = 1; a couple of times per year: n = 3) and for private purposes only. The used ride-pooling services were Moia (n = 3), UberPool (n = 2), and CleverShuttle (n = 2).

The group of participants was familiar with pedestrian navigation systems via smartphones, such as Google Maps (96%). A total of 21 participants used such systems a couple of times per year, 12 stated monthly, and 7 (weekly: n = 6; daily: n = 1) even more frequent use thereof. The majority of the participants (n = 28) experienced mixed reality techniques (AR = 86%; VR = 54%) before. The overall technology affinity of the study participants was slightly below average (ATI score = 2.81; SD = 0.77; on a 6-point Likert scale, 1 = "do not agree" to 6 = "fully agree") [45]. The study was conceptualized and realized in accordance with the Declaration of Helsinki. Informed consent was obtained from all participants before the experiment. The participants were allowed to stop the study at any point without justification or consequence. The participants volunteered but were financially compensated with EUR 5 per 30 min.

3.1.2. Dependent Variables and Questionnaires

The HMI prototype information elements were evaluated regarding *comprehensibility* during a naïve run using a 7-point Likert scale (1 = "not comprehensible at all"; 7 = "very comprehensible") to check whether participants understood the AR interface.

*User experience* was measured using the standardized User Experience Questionnaire with 26 items of contrasting attributes on 7-point scales [24].

*Workload* was measured using a modified and shorter version of the standardized NASA-TLX questionnaire, focusing only on the TLX items. Pairwise comparisons of items were left out, and the scope of the scales ranged from 1 = "very low demand" to 10 = "very high demand" [46].

The UTAUT2 questionnaire was modified to measure the *acceptance* of the HMI prototype. Only two subscales ("usability" and "intention to use"; 3 items each) were taken. Answers were recorded on a 7-point Likert scale (1 = "don't agree at all"; 7 = "totally agree") [47].

Additionally, forced multiple choice questions were used asking the participants in which circumstances they would most likely use the prototype, if available. Possible answers were the following: *in familiar territory*, *in unfamiliar territory*, *for long routes*, *for short routes*, *in buildings*, *on the open street*, *under time pressure*, *for initial orientation*, *for the final location*, and *other* with an open input box.

3.1.3. Materials

Following the results of the first user study, an AR high-fidelity prototype for smartphones was developed based on Google's AR-Core framework for Android. Common AR design and usability heuristics were applied [48–50]. To implement the information strategies for all scenarios in a feasible manner, the information elements were accordingly adapted. By this approach, we accounted for the limitations of the used AR framework during prototype development. For example, the *path* element was abstracted using a "breadcrumb" analogy. Hence, the continuous path was divided in small consecutively arranged parts that would still show a pathway. Abstraction was chosen to make the *path* information as intuitive to understand as possible. Otherwise, the whole way until the vStop would appear on the smartphone right away, corners or buildings would be misunderstood (due to lack of depth sensor), as well as data handling and processing would be too challenging for the system. The successively appearing breadcrumbs were of circular shape, containing a footstep icon and leading the way to the pickup spot (see Table 3). The next five elements were constantly shown. The vStop element was realized by showing a *standing subject* as an augmented flag with the German bus-stop sign (VZ 224 StVO). It also showed a round-shaped *waiting area* on the *floor*, which showed the writing "your waiting area" (see Table 3). The *standing subject* and *waiting area* elements would appear as the final breadcrumb to end the navigation. All elements to digitally augment the

environment were of light blue color tone and semitranslucent in order not to completely cover the ground. For the round-shaped elements placed on the ground (*waiting area* and *path*), cyan-colored outlines were added to increase contrast and improve recognizability. The color scheme was chosen to give a neutral look and ensure good visibility due to its contrasting appearance (color tones and environment). In addition, for color-sensitive users, blue and cyan easily separate from the rest of the environment on the smartphone screen. The *orientation* arrow was held in dark gray in front of a light-gray box to make sure that it is sufficiently recognizable on the smartphone screen and among the other objects (see Table 3). The arrow would always point toward the next information element (e.g., *path* element). The *orientation* element would turn red if the smartphone was not held into the right direction and the next *path* element was not in sight, so the user would immediately recognize it. Figure 2 shows the arrangement of all elements in the mobile AR interface. The vehicle indicating an AR element was not realized in the prototype due to technical limitations. Accordingly, the prototype only spanned the first two scenarios of the user journey (navigation and identification of the vStop).

**Table 3.** vStop information elements of AR HMI prototype.

| vStop Element | |
| --- | --- |
|  | <ul><li>Pole standing upright</li><li>Flag with German stop sign (VZ 224 StVO)</li><li>Round-shaped waiting area on the ground</li><li>"Your waiting area" written on floor element</li><li>Final element of navigation</li><li>Shows where pickup position of vehicle will be</li></ul> |
| **Pathway Element** | |
|  | <ul><li>Many breadcrumb-like elements to mimic pathway</li><li>With footstep icons in walking direction</li><li>Consecutively placed on the ground</li><li>Shows user the route to walk</li><li>Always the next 5 path elements are shown, vanish behind user</li></ul> |
| **Orientation Element** | |
|  | <ul><li>Arrow overlay on smartphone screen</li><li>Points toward next information element (path or vStop)</li><li>Rotates in 22.5° steps</li><li>Changes color to red if next information element is not in sight</li></ul> |

### 3.1.4. Study Design

The goal of the study was to evaluate the vStop HMI regarding user experience, acceptance, and workload. Common usability testing heuristics for mobile AR were addressed [51].

The participants had to solve the task of navigating under time pressure to a fictive vStop location in unfamiliar territory. The only assistance they had was the HMI prototype. The setting was chosen because the first user study showed slightly higher user requirements for information presentation in unfamiliar territory than in familiar territory. Hence, the developed HMI should prevail in field test conditions of high user information requirements.

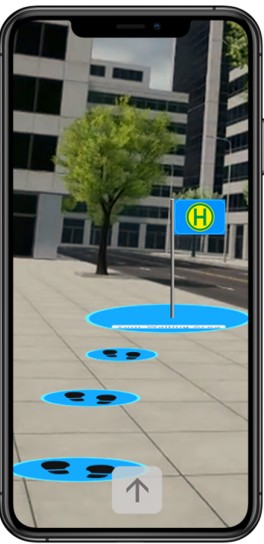

**Figure 2.** Exemplary arrangement of information elements in vStop HMI prototype.

The route to navigate was approximately 70 m long and led from an inner courtyard through a building to a main road where a SAV pickup could take place (see Figure 3). The length of the route can be described as somewhat realistic. The route was not marked by any physical cues, and it was ensured that no disturbance, such as other persons, would come across the route during testing. Although there was no SAV available for the study, the participants were told to imagine that this could be a real SAMOD scenario.

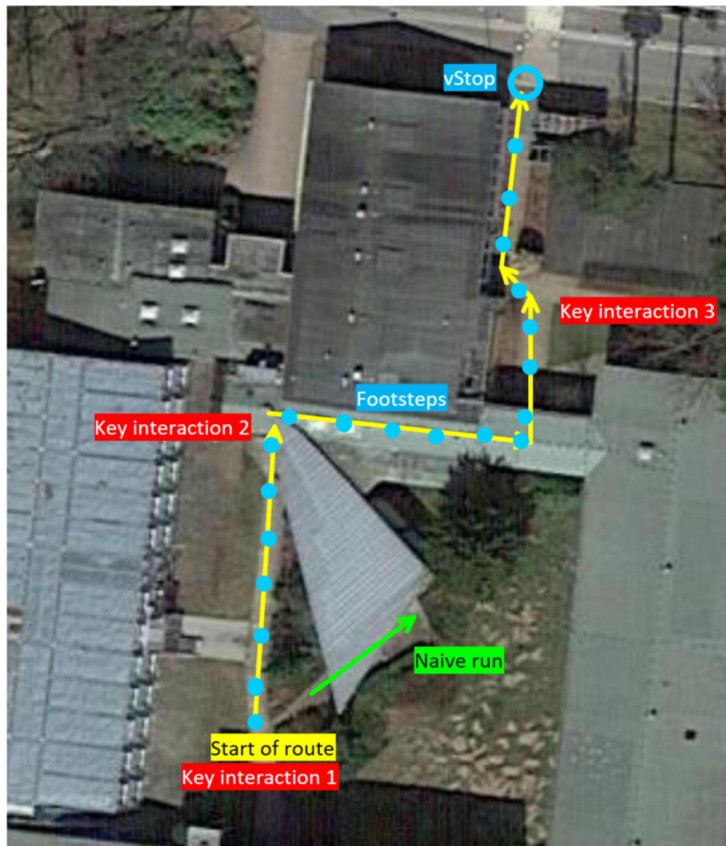

**Figure 3.** Overview of the test area of second user study. Green: naïve run route; yellow: field study navigation route; blue: AR-prototype information elements (path breadcrumbs and vStop); and red: situations of key interaction with the HMI (taken from Google Maps with own markings).

AR-pathway elements were placed along the way in a distance of approximately 5 m each, with the vStop element as the last element. Due to technical restrictions with the AR application, the route was manually prepared prior to every trial. Nevertheless, smartphones would still quickly heat up, and the application was very sensible to crashing. We followed a standardized procedure to ensure the consistency of the virtual and physical conditions. In the case of crashing of the app, the instructors took over and recalibrated the app and the trial would continue.

There were three key interactions of particular interest alongside the route in which the performance of the interface would be crucial. The key interactions were chosen to assess the participants' interaction with the novel AR prototype in a realistic situation in detail (see Figure 4). The key interactions were as follows. The first was the initial orientation, in which the participants started with their back orientated toward the route so they had to figure out their own orientation first. The participants had to correctly interpret the arrow symbol and find the first pathway elements that would lead the way. The second key interaction was right after entering the building, where the route took a sharp 90-degree right turn behind a door. Again, the participants had to correctly understand the HMI behavior; otherwise, they would end up walking a wrong way. The last key interaction was right before arriving at the pickup location. The users would see the vStop element for the first time on the smartphone, but the route took a little swerve so the interface would place the vStop element in the bushes for a short moment, which could lead to confusion.

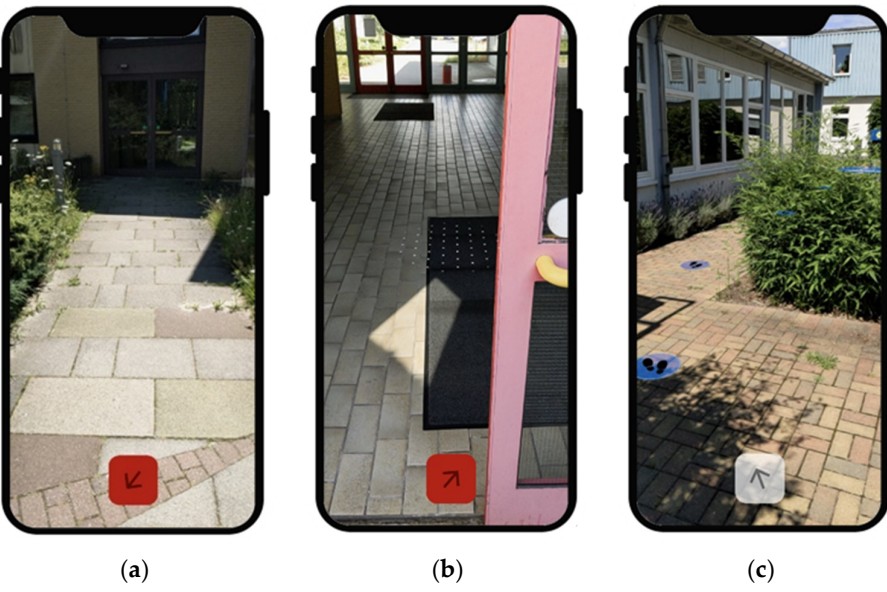

**Figure 4.** View of smartphone screen showing AR prototype during field study: (**a**) key interaction 1 shows interface during initial orientation; (**b**) key interaction 2 shows interface when entering building, followed by a sharp turn right; and (**c**) key interaction 3 shows interface when vStop was in sight for the first time and information elements look like partly placed off route.

### 3.1.5. Procedure

The user study to evaluate the vStop AR prototype in the field was conducted at the DLR Campus in Braunschweig, Germany. The whole study procedure followed an approved COVID-19 safety protocol and was mainly conducted outdoor. All participants were tested for COVID-19, and all involved personnel had to wear masks. Throughout the study, the instructors checked for distancing and disinfection of mobile devices.

The study was structured as follows: In the first part of three, participants were picked up at the gates by the researchers and were guided to the facilities where the study would take place. The researchers made sure that the participants would not see the test environment beforehand. Participants were introduced to the research topic, and a naïve run was conducted to explain the AR prototype in detail. The prototype was

presented in a test environment with an approximately 10 m long route. All information elements and the functionality of the prototype were presented. The participants were given the smartphone (Google Pixel 4a), and they could freely discover how the AR interface would behave; all the remaining questions were answered. During the naïve run, the participants got familiar with all the functionalities of the HMI and were informed that crashing of the app (if occurred) was not part of the study. After getting to know the prototype, the participants were asked to evaluate each information element of the interface regarding comprehensibility.

The second and main part of the user study followed, and the participants were asked to put themselves into the situation of using a SAV on-demand. The task was to navigate to the pickup location within a given timeframe (3 min) only by utilizing the AR prototype. The task was completed when the participants believed to have reached the vStop (see Figure 5).

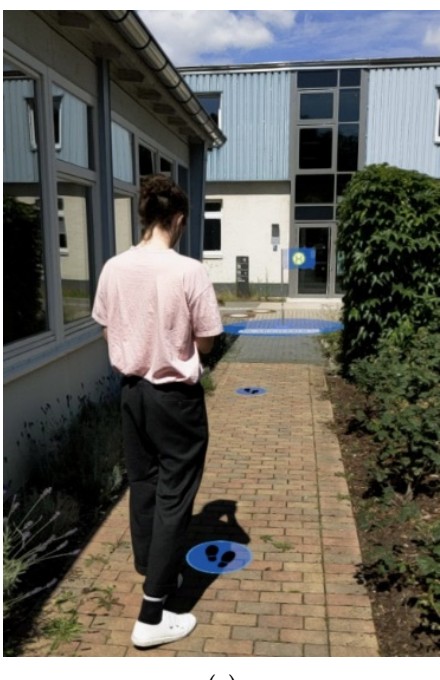 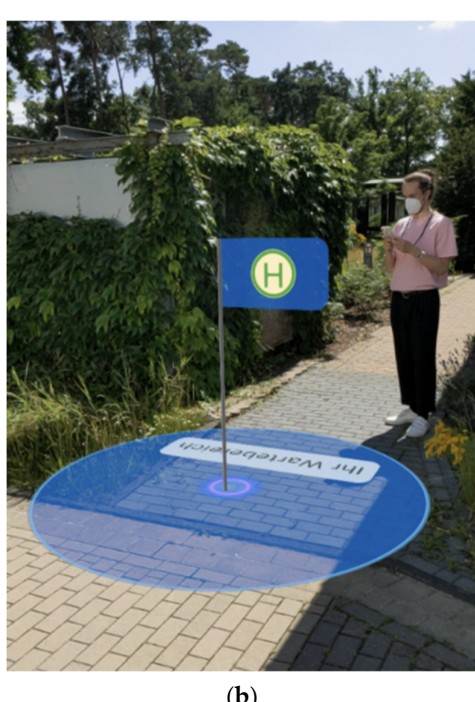

(**a**)        (**b**)

**Figure 5.** Participant during AR testing in second user study. Photo arrangements of AR functionality for representational reasons. (**a**) Participant follows pathway elements in order to navigate toward vStop. (**b**) Participant has reached vStop and is positioned right before the labeled waiting area.

During the navigation task, an instructor followed the participants to make sure to intervene in case of any unforeseen app crashes or complications with the interface. The following distance was at about 3 m to still be able to check what the participants would see on the smartphone screen while also not interfering with the task or giving hints to the participants in which way to go.

The user study ended with filling of a set of questionnaires and a short structured interview with open questions about what the users found positive or negative about the prototype and suggestions in terms of improvements they would like to see. They were asked to evaluate the HMI prototype with respect to its flaws by tolerating its early stage character. The goal of the interview was to gain qualitative understanding about the prototype from a user perspective.

### 3.2. Results

The next paragraph thoroughly presents the results of the field study. First, data analysis is described. Second, the results of the naïve run that consist of comprehensibility ratings of the vStop prototype are depicted. Third, the results of the standardized questionnaires

UEQ, modified NASA-TLX, and modified UTAUT2 are presented. Finally, the subsection shows the participants' qualitative feedback regarding the vStop HMIs' preferred situations of use, likes, dislikes, and potential improvements.

### 3.2.1. Data Analysis

In the beginning of the data analysis, outliers were detected according to the standardized UEQ guideline procedure. Accordingly, a participant is to be removed when it can be suggested that he or she did not seriously answer all items. The criterion for exclusion is quantified by showing three UEQ subscales with an answer distribution of an item >3 (on a seven-point Likert scale) between the best and worst evaluations. One critical response was detected and deleted as an outlier. The outlier was deleted across all questionnaires, except the naïve run. Moreover, no critical answer distributions were detected across all questionnaires.

Cronbach's alpha analysis of UEQ subscales resulted in high internal consistency among all scales. Novelty ($\alpha = 0.83$, four items), stimulation ($\alpha = 0.84$, four items), perspicuity ($\alpha = 0.90$, four items), and attractiveness ($\alpha = 0.92$, six items) showed high to very high alpha values. For dependability ($\alpha = 0.68$, four items) and efficiency ($\alpha = 0.77$, four items), the alpha values were at acceptable levels. Hence, the UEQ data were suitable for interpretation. The same was applied for the interpretability of the modified NASA-TLX ($\alpha = 0.76$, six items) questionnaire. The modified UTAUT2 questionnaire with the chosen subscales usability ($\alpha = 0.94$, three items) and intention to use ($\alpha = 0.91$, three items) also showed very high rates of internal consistency.

The data were analyzed and showed a nonparametric distribution over all dependent variables. To evaluate a significant impact of the prototype, a Wilcoxon ranked sum test was conducted for each set of questionnaires.

The overall user experience was measured by the UEQ and was divided into six subscales, which can be attributed to three separate dimensions (attractiveness, pragmatic quality, and hedonic quality). High ratings in attractiveness indicate that the prototype was likeable on the first impression. Perspicuity, efficiency, and dependability account for the prototype's goal orientation (pragmatic quality). If it was easy for users to get familiar with the prototype, perspicuity ratings are positive. High values in efficiency indicate that the users were able to easily solve their tasks. If the users feel in control during the interaction with the prototype, dependability values are high. Hedonic quality accounts for the user's joy of use with the prototype and is reflected in high stimulation (is it exciting?) and novelty (is it creative?) ratings. All subscales were modified post hoc and range from −3 = "horribly bad" to 3 = "extremely good". In order to have positive evaluation, the mean values of the pragmatic quality, hedonic quality, and attractiveness subscales ought to be significantly higher than 0.80.

The NASA-TLX questionnaire consists of three items that measure physical, mental, and temporal demand while using the prototype. Additionally, three items measuring performance, frustration, and effort for solving the task are included. The threshold for low workload while using the prototype was set by the researchers to reach mean values significantly lower than 5.00 (on a 10-point scale).

The overall acceptance of the HMI prototype was measured using a modified version of the UTAUT2 questionnaire. The usability scale consisted of items measuring enjoyment, fun, and entertaining aspects during the interaction with the HMI. Future intentions to routinely use the prototype and frequent usage of the prototype go with the intention-to-use scale. Acceptance is considered high when the mean values of both subscales as well as the overall UTAUT2 mean values are significantly higher than 4.00 (on a seven-point scale). This threshold was also set by the researchers.

### 3.2.2. Naïve Run

The overall prototype interface and all single information elements were rated very comprehensible by the participants (all descriptively showing values above 6.00). It is

important to mention that only participants who correctly identified the element accounted for the comprehensibility score (Table 4). Almost all elements' meanings were correctly identified by the participants. Only one participant thought that the arrow for orientation would always point toward the pickup location. Two participants were irritated by the breadcrumb element because they misunderstood the footstep icon as to stand still at this position. Both misinterpretations reflected the relatively high standard deviations of the elements (SD = 1.34 for orientation; SD = 1.28 for pathway). In total, for the participants, the HMI was very easy to understand (M = 6.56; SD = 0.83).

**Table 4.** Results of AR-element comprehensibility rating by participants of second user study during naïve run.

| HMI Information Element | Comprehensibility M (SD) | n * |
|---|---|---|
| Interface as whole | 6.56 (0.83) | 45 |
| Arrow (orientation) | 6.27 (1.34) | 44 |
| Footstep breadcrumbs (pathway) | 6.56 (1.28) | 43 |
| Waiting area (circle) | 6.91 (0.28) | 45 |
| "Your waiting area" (writing) | 6.82 (0.68) | 45 |
| Flag (vStop indicator) | 6.87 (0.34) | 45 |
| "H" (stop sign) | 6.96 (0.29) | 45 |

* n describes the number of participants who correctly interpreted the element.

### 3.2.3. User Experience

The UEQ results show that the vStop HMI was received significantly positive on the first impression by the users (attractiveness: M = 1.85; SD = 0.92, $p < 0.001$). Pragmatic quality (M = 1.99; SD = 0.76) was very high, which indicates that the prototype was very helpful to solve the given task. Table 5 shows that all contributing subscales (perspicuity, efficiency, and dependability) are significantly positive at the $p < 0.001$ level. Hedonic quality (M = 1.30; SD = 1.01) was also evaluated to be significantly positive at the $p < 0.001$ level and indicates that users enjoyed using the prototype. Subscale stimulation was evaluated to be significantly positive at the 0.001 level. Only the subscale novelty was not evaluated to be significantly better than 0.80 ($p = 0.09$).

**Table 5.** UEQ ratings of vStop HMI prototype by participants of the second user study (N = 44).

| UEQ Scales | $M_{emp}$ ($SD_{emp}$) | 95% CI [LL;UL] [1] | $M_{crit}$ | V [2] | $p$-Value$_{est.}$ [3] |
|---|---|---|---|---|---|
| Novelty | 1.05 (1.21) | 0.69; 1.41 | >0.80 | 608 | 0.09 |
| Stimulation | 1.54 (0.95) | 1.26; 1.82 | >0.80 | 846 | <0.001 |
| Hedonic Quality | 1.30 (1.01) | 0.99; 1.60 | >0.80 | 766 | <0.001 |
| Dependability | 1.88 (0.80) | 1.64; 2.11 | >0.80 | 950 | <0.001 |
| Efficiency | 1.79 (0.96) | 1.51; 2.07 | >0.80 | 899 | <0.001 |
| Perspicuity | 2.31 (0.77) | 2.08; 2.54 | >0.80 | 979 | <0.001 |
| Pragmatic Quality | 1.99 (0.76) | 1.76; 2.22 | >0.80 | 969 | <0.001 |
| Attractiveness | 1.85 (0.92) | 1.57; 2.12 | >0.80 | 927 | <0.001 |

[1] Confidence interval with lower (LL) and upper limits (UL); [2] test statistic that indicates the sum of positive rank numbers; [3] parameter $p$ can only be estimated since ties exist in the data.

### 3.2.4. Workload

The overall TLX score indicates that the vStop prototype was very supportive to the users, which led to very low rates of workload while solving the given task. The significantly low value of the total TLX score (M = 2.27; SD = 1.15; $p < 0.001$) and all six associated items highlights the assisting character of the prototype (see Table 6).

**Table 6.** NASA-TLX ratings of vStop HMI prototype by participants of the second user study (N = 44).

| NASA-TLX Items | $M_{emp}$ ($SD_{emp}$) | 95% CI [LL,UL] [1] | $M_{crit}$ | V [2] | $p$-Value$_{est.}$ [3] |
|---|---|---|---|---|---|
| Mental demand | 2.50 (1.61) | 2.01; 2.99 | <5.00 | 44.5 | <0.001 |
| Physical demand | 1.84 (1.48) | 1.39; 2.29 | <5.00 | 14 | <0.001 |
| Temporal demand | 2.39 (1.73) | 1.86; 2.91 | <5.00 | 42.5 | <0.001 |
| Performance | 2.21 (1.75) | 1.67; 2.74 | <5.00 | 53 | <0.001 |
| Effort | 2.36 (1.86) | 1.80; 2.93 | <5.00 | 49 | <0.001 |
| Frustration level | 2.30 (1.80) | 1.74; 2.85 | <5.00 | 18 | <0.001 |
| TXL modified total | 2.27 (1.15) | 1.92; 2.61 | <5.00 | 5.5 | <0.001 |

[1] Confidence interval with lower (LL) and upper limits (UL); [2] test statistic that indicates the sum of positive rank numbers; [3] parameter $p$ can only be estimated since ties exist in the data.

### 3.2.5. Acceptance

Focusing on the usability and intention-to-use dimensions of UTAUT2, the overall score indicates a significantly positive acceptance rate (M = 5.03, SD = 1.49; $p < 0.001$). In addition, both subscales usability (M = 5.10; SD = 1.59; $p < 0.001$) and intention to use (M = 4.96; SD = 1.60; $p < 0.001$) were evaluated to be significantly positive. Only item I2 ("daily use") did not reach the threshold for significant positive evaluation (see Table 7).

**Table 7.** Modified UTAUT2 ratings of vStop HMI prototype by participants of the second user study (N = 44).

| UTAUT2 Items and Scales | $M_{emp}$ ($SD_{emp}$) | 95% CI [LL,UL] [1] | $M_{crit}$ | V [2] | $p$-Value$_{est.}$ [3] |
|---|---|---|---|---|---|
| Enjoyable (U1) | 5.10 (1.71) | 4.57; 5.61 | >4.00 | 605 | <0.001 |
| Fun (U2) | 5.43 (1.68) | 4.92; 5.94 | >4.00 | 745 | <0.001 |
| Entertaining (U3) | 4.75 (1.63) | 4.25; 5.26 | >4.00 | 528 | <0.005 |
| Usability total | 5.10 (1.59) | 4.61; 5.57 | >4.00 | 779.5 | <0.001 |
| Intention to use in future (I1) | 5.55 (1.52) | 5.08; 6.01 | >4.00 | 744 | <0.001 |
| Always try to use in daily life (I2) | 4.36 (1.94) | 3.77; 4.95 | >4.00 | 428 | 0.12 |
| Continue to frequently use (I3) | 4.98 (1.70) | 4.46; 5.50 | >4.00 | 589 | <0.001 |
| Intention to Use total | 4.96 (1.60) | 4.48; 5.45 | >4.00 | 696.5 | <0.001 |
| UTAUT2 modified total | 5.03 (1.49) | 4.57; 5.48 | >4.00 | 784 | <0.001 |

[1] Confidence interval with lower (LL) and upper limits (UL); [2] test statistic that indicates the sum of positive rank numbers; [3] parameter $p$ can only be estimated since ties and zeros exist in the data.

### 3.2.6. Participants Preferred Situations of vStop HMI Usage

The participants were asked in what circumstances and situations they would use the vStop prototype, if available. For this question, a forced multiple-choice set of answers was given.

Unanimously, all participants stated that they would preferably use the vStop HMI in unfamiliar territory. In contrast, the HMI was perceived as not very helpful in familiar territory (n = 5). Seventy-three percent of the participants would use the HMI especially indoor, and 56% would use it outdoor. In addition, high rates of potential usage when being under time pressure (73%) were stated by the participants as they desire efficient navigation to their pickup location. A total of 19 participants would prefer the HMI for rather long routes to the vStop and 22 for rather short routes. A very large number of participants (n = 31) favored the AR solution in particular for identifying the exact pickup location. This shows the importance of the vStop element at the end of the navigation task. Twenty participants would like to use the HMI for initial orientation at the beginning of the navigation task. Other circumstances openly mentioned by the participants were "at night" (n = 1), "during emergency situations as escape routing" (n = 1), and "at airports" (n = 1).

3.2.7. Qualitative User Feedback Regarding Likes, Dislikes, and Potential Improvements of the vStop HMI

At the end of the study, the participants had the possibility to give remarks in a qualitative way with regard to the experienced prototype. Fourteen participants highlighted very good overall comprehensiveness of the prototype. The presented elements (n = 7) and AR technology (n = 3) were also highlighted to be very helpful. The most appreciated AR information elements were the footstep breadcrumbs to show the pathway (n = 22) and the arrow element (n = 19). The arrow's color adaptation and rotation according to orientation was perceived as helpful. The flag element to show the exact pickup location was also mentioned to be easy to recognize, even from a distance (n = 11). Other aspects such as the use of pictograms (n = 1) and color of elements (n = 1) were mentioned to be positive, too.

Although, only two participants mentioned that they felt unaccustomed to using AR for the navigation task, five participants did not fancy that they were very focused on the smartphone screen during the task completion. Sixteen participants mentioned that the prototype was unclear especially in the second key interaction (entering the building, followed by a sharp right turn) caused by the lack of information provided. In two cases, the participants noted that the AR elements had shifted (pathway icons were placed in the bushes), and two participants criticized the distance of the footstep icons as being too far away from each other. Apart from that, four participants mentioned irritations caused by the behavior of the orientation arrow, as it was a 2D overlay with nonsteady behavior. The arrow seemed very hard to interpret when the phone was not held upright.

The most desired improvement of the prototype mentioned was more information at the second key interaction, when entering the building and turning right. Fifteen participants articulated the need for better behavior of the orientation element or an additional information element. In general, six participants would have liked to see a continuous path element instead of the footstep breadcrumbs, and three participants asked for more dense distribution of breadcrumbs, which they mentioned to having had to search for (n = 2). Six participants would have liked to see a conventional map in addition to AR. For the orientation element, a different three-stage color coding (green, yellow, and red) was suggested (n = 2). The 3D positioning of the arrow (n = 1) and dynamic size adaption, depending on the distance to the next turning point (n = 1), were also mentioned to be desirable. Five participants suggested that the prototype could have given visual or haptic feedback, when arriving at the actual vStop element, so that one would instantly recognize that the task was correctly solved. One participant also asked for an element indicating the start of the navigation (n = 1). Other additionally desired information elements were a signal to enter the building (n = 2), warning when being at the street (n = 1), and a compass-like element always pointing to the vStop (n = 2). Other information methods such as acoustic (n = 2) and haptic (n = 2) signals when getting off the track where also mentioned to add to the prototype.

Overall, the vStop HMI was very well received by the participants and provided a positive experience. The targeted problem was highly relatable to the participants, and the prototype helped in solving the task to their satisfaction. None of the participant mentioned disliking or aversion toward the HMI.

## 4. Discussion

In two consecutive user studies, this work focused on the HMI aspects of the vStop concept for SAMOD. The focal point of this research was set on the navigation scenario and identification of pickup location in the SAMOD user journey.

The first user study was conducted as an online interview study to identify user preferences for AR information entities. Thereafter, the field study had the goal of evaluating an HMI prototype, which consisted of these AR information entities.

In a nutshell, the results of both conducted studies are complementary and contribute to an efficient vStop HMI giving guidance to users along the SAMOD user journey. The AR information entities identified in the first interview study by the participants were

transferable to a vStop prototype, which was evaluated as being very supportive by the participants in the second user study. The prototype was very well accepted overall, and means of AR proved to be a worthy solution for the given task of navigating through unfamiliar territory and identifying the specific pickup spot. For the investigated use case of efficiently identifying a vStop, the results indicate that the AR HMI concept is likely to be a helpful and desirable solution from a user perspective.

The participants' evaluation of AR information element arrangements in the first study showed that the users would rather prefer few but meaningful elements on a mobile AR interface. This finding is in line with mobile AR design heuristic to avoid cluttering [50]. In scenarios 1 (navigating) and 3 (identification of the pickup vehicle), the highest possible amount of information elements even reached to the lowest usability ratings among all possible information element combinations. In addition, none of the maximum information element interfaces was chosen as favorite.

The users' choices of favored information element arrangements were highly consistent for unfamiliar and familiar territory. Interestingly, only one interface option that received the highest usability score was also chosen as the participants' favorite (the interface only showing a path in the first scenario) and only so for familiar territory.

In all three investigated scenarios in the first user study, discrepancies between the usability score and favored choices were found. In scenario 1, the favorite interface for unfamiliar territory and the third most popular for familiar territory (*path* and *turning node*) did not even reach the >6.00 threshold. In scenario 2, the combination of *orientation*, *standing*, and *floor* was among the top 3 favorites in familiar and unfamiliar territory but only showed a usability score of 5.80. However, in contrast, the interface with the highest usability score (showing just a *standing* element) was not even ranked in the top 3 for favorite interface in unfamiliar territory (but then the second most popular in familiar territory). Inconsistencies were also found in scenario 3 in which an interface that showed a mean usability score of 5.62 (*object + stopping area*) was ranked the second most popular in both circumstances. Although all the results of the first user study were not significant, it can be said that, among the users, only a few favorite AR interfaces really stood out. For example, in scenarios 1 and 2, the users expressed the desire to add the *orientation* element when choosing the favored interface. On the one hand, this indicates that the participants were able to transfer the shown mock-up pictures to a real-use scenario in the first study. On the other hand, when in doubt, the users might ask for more information, and despite the new possibilities of AR, this shows that a conventional and familiar compass-like information element would still be desired.

The option for a *floating* element was also inconsistently evaluated by the participants of the first study. For example, in scenario 2, the single *floating* element showing the pickup location was evaluated relatively low in usability across all categories compared with the other presented interfaces. Still, it received 14% of the popular vote for familiar territory. This indicates that participants liked the idea of showing the vStop with this information strategy. However, they obviously were not satisfied with the quality of information that this information entity provided.

The first study was successful in identifying high-quality information elements. The results provided a very solid foundation for the next step of developing a vStop HMI. In preparation for the second user study, all information elements were successfully implemented in accordance to technical conditions and AR design heuristics. The choice of the final interface was characterized by limitations but eventually led to a supportive HMI.

The effectiveness of the prototype is underlined by the significantly positive rates of UX, workload, and acceptance by the participants of the field study. Accordingly, all investigated dependent variables were in accordance with each other, and no conflicting results were detected. As for the UEQ rating, significantly positive pragmatic quality clearly stood out. Especially, the subscales dependability, efficiency, and perspicuity showed significantly high positive scores and proved the usefulness of the HMI. This means that the mobile AR information system provided a high ease of use and was very easy to understand

and predictable in its behavior. Hence, the prototype was very helpful when the users navigated through unfamiliar territory. These results were also reflected in the usability subscale of the modified UTAUT2 questionnaire, which also showed significantly positive results. The prototype's assistive character was also reflected in the significantly low ratings of workload during the task completion. In these aspects, the findings correspond with previous research regarding the use of AR [32,37,38].

The stimulation subscale of the UEQ showed significantly positive evaluation and indicates that participants also enjoyed using the prototype. This outcome adds to former findings that AR can promote hedonic quality during the navigation tasks [41].

Additionally, a significantly high score in attractiveness underlined the positive attitude of the users toward the vStop HMI. Overall, the HMI demonstrated a very high potential to be an effective source of information in the investigated use case. In combination with a positive acceptance rating, which was reflected in significantly positive scores in the intention-to-use subscale of the UTAUT2 questionnaire, the concept of a mobile AR HMI proved to be a handy, useful, and desirable solution for users of SAMOD in the future.

In terms of qualitative user feedback, valuable insights were gained particularly about how users would interact with a mobile AR HMI. The chosen research approach demonstrated that the users have different perceptions about how the AR information representation would be most helpful to them when really experiencing the HMI. For example, the three key interactions in the real-life exposure user study revealed that the desired information supply could be higher in specific situations. The differentiation between familiar and unfamiliar territory in the first user study already tried to address this aspect. Nevertheless, conducting two user studies to develop and evaluate an HMI in one joint work proved to be key for this finding. Future HMIs could be responsive depending on the situation, circumstance, and/or user itself. The feedback was important to identify the most pressing areas of improvement of the HMI, such as the orientation and clear indication of beginning and end of navigation to provide better reliability of the displayed information. However, in the conducted field study, the participants experienced the prototype for the first time and successfully navigated to a fictive vStop under time constraints, only receiving information via a mobile HMI.

The field study showed that the initial interview study to narrow down the AR elements resulted in a highly acceptable and efficient interface of low workload during the task. During the field study, all AR elements received exceptionally well comprehensibility ratings. Hence, the online interview study results were confirmed under real-life circumstances. The prototype was easy to use for the participants, and for this specific use case, AR technology can be seen as a suitable modality.

The results of this work provide a very solid base for further iterating the vStop HMI. The chosen approach to develop the HMI with a strictly user-centered focus proved to be successful in the field study, although the users were only presented with static sketches in the interview study. The field study also delivered valuable insights about how users would actually interact with the mobile AR HMI because, in part, the participants showed different using behaviors with the first tangible prototype. Although they were familiar with smartphones and despite the in-depth explanation of the prototype during the naïve run, some participants were holding the smartphone parallel to the ground like a digital map. This, for example, led to uncertainty while interpreting the arrow element instead of following the footstep breadcrumb icons. The interface was easier to interpret and use when it was pointed forward in walking direction (just like taking a picture). The majority of the participants were able to use the interface as intended, which, on the other hand, resulted in the participants being very focused on the smartphone while walking. Nevertheless, usage behavior such as this can be seen as highly valuable for future HMI iterations. Additionally, investigation of the key interactions proved to be valuable in identifying the shortcomings of the developed HMI prototype in terms of information gaps along the user journey. Overall, the findings point out that an HMI prototype already meets the user information needs very well.

### 4.1. Limitations

Apart from the successful design and development of the HMI, both user studies showed limitations. In the first study, the user group was relatively small but very focused (young adults) to obtain coherent feedback for design requirements. The investigated interview scenarios were of hypothetical nature and visually supported only by static pictures. HMI visualizations were also static pictures only. Eventually, the choices of the best information elements of the HMI were made by own descriptive criteria. Future online studies can integrate video material of how the interface will behave in each situation to foster immersion during the interview. Feedback for interface behavior was then only gained in the real-life exposure study. In the field study, the user group was acceptable in size and relatively diverse in terms of age and background, not like the very focused user group in the first study. However, the results were still very consistent. Although the experimental character of the second user study lessened the internal reliability, the validity was at an acceptable level because the setting, task, and stimuli were controlled as much as possible. The AR functionality was very limited, and the app was prone to crashing or screen freezing. Careful smartphone handling and trial setup had to be ensured. For example, in one case, the interface was malfunctioning and the arrow symbol froze. The trial was reconducted without further troubles. The route had to be manually set up for every trial, which might have resulted in small inconsistencies during the AR-element placement. To overcome these technical issues, the AR prototype would have needed more extensive durability and reliability testing as well as detailed bug fixing to provide more consistent performance.

The investigated scenario in study 2 was cut down to the essential tasks of navigating and identifying the pickup location. In this study, the focus was set on a first encounter with the AR prototype in a restricted setting. Neither were users able to go through a booking process and the smartphone was directly handed over for navigation nor was there a real pickup vehicle involved. Both aspects should be taken into consideration in future studies to gain understanding about the users' information needs and pain points and to improve the external reliability. As all participants were able to easily get to the final location in time, the countdown (3 min) might have been tighter to put more pressure on the participants, or the situation or test area could be more complex.

### 4.2. Further Research

Proceeding from this work, further research derives to investigate the performance of the vStop HMI and to iterate the AR prototype. Especially, the user interaction with the mobile AR interface needs more thorough research from a human-factor perspective. Therefore, testing in realistic environments to address the functionality in boundary conditions of using the HMI should be targeted. Aspects such as spatial knowledge, situation awareness, attention, and user circumstances (fatigue and stress) should be investigated in the future. In addition, interface design aspects such as gamification, aesthetics of AR elements, use of color and contrasts, and other interaction design heuristics for AR should be researched onward. Measures to improve the hedonic quality of the prototype can focus on iterating the actual design (meaning styling) of and interaction with the AR elements, which was not the focus of this research. In addition, differences in the amount of AR information elements in unfamiliar and familiar territory should be investigated further in more naturalistic studies. Furthermore, the usability of the HMI should be investigated in daily-use scenarios over a long time period to evaluate the truly essential AR information elements for this use case.

The technical modality for the present vStop HMI (AR) should be tested against other modes of navigation (voice, digital maps). In addition, other AR modalities (wearables, glasses) and other concepts of AR (landmark-based) for navigation tasks should be investigated, too. In that regard, the user journey needs further investigation in order to understand which sequences are supported best by which information modality. In addition, the multimodality of a vStop HMI should be addressed.

In future research, the pickup scenario should be addressed, too, in order to gain understanding about identifying the SAV. After all, the matching capability for the user of the SAV by the HMI was not investigated but remains a key interaction of the vStop HMI. Additionally, the HMI interaction with further traffic infrastructure elements needs to be added to the scenarios. This will also contribute to more holistic vStop capabilities and AR-interface use cases, which are very likely to be relevant in future SAMOD deployment. This will also contribute to even more helpful HMI design, for example, by implementing information about other road users in the vicinity or a potential ad hoc shift of the vStop location due to conditions in the local traffic space. The traffic controlling and superordinate virtual infrastructure element perspective of vStops was left out in this research. Prospectively, the display of not only service-related information but also real-time data about the traffic environment should also be considered in next interface designs. Building on this work, the researchers will go into the next iteration phase of the user-centered design process to designing an efficient and assisting vStop HMI.

## 5. Conclusions

In developing a novel vStop HMI, this paper delivered the first insights and an innovative step toward user-centered design of virtual traffic infrastructure by means of mobile AR. This present work was based on a twofold user study approach and presented successful design and development of a first AR vStop HMI for mobile devices. Firstly, a vStop HMI prototype was designed by means of AR. Therefore, an online interview study was conducted, presenting various potential AR interfaces of different information quantities to users. Interfaces were evaluated regarding usability and personal preferences for a SAMOD use case. Secondly, the AR prototype was developed for mobile AR and eventually tested regarding UX, acceptance, and workload in a real-life exposure field test.

In a nutshell, the design, development, and testing of the vStop HMI were very successful. By conducting two user studies, the researchers were able to successfully design a prototypical and highly supportive information system, which could be brought to the user's hands in the future. Consequently, this work delivered valuable insights for the field of user-centered virtual traffic infrastructure, mobile AR design, and AR pedestrian navigation. With the vStop HMI, an overall high user experience of SAMOD can be assured in future pickup scenarios, which can lead to high adoption rates of automated on-demand shuttle services. In the field study, all the participants were able to find the pickup spot in time and with little effort. In conclusion, mobile AR proved to be a very efficient HMI modality to assist users with meaningful information along the first scenarios of the SAMOD user journey (navigation and identification of a specific position). The interface design and information elements proved to be very helpful in the investigated scenario. The vStop prototype serves as a sound starting point for further iterations in research of user-centered vStop HMI design. In the future, SAMOD service providers can equip their applications with vStop HMIs so that customers can seamlessly access their automated ride. With this approach, user acceptance, smooth operation of the service, and change toward more sustainable modal choices of individuals can be fostered.

**Author Contributions:** Conceptualization, F.H. and M.O.; methodology, F.H. and M.O.; formal analysis, F.H.; data curation, F.H.; writing—original draft preparation, F.H. and M.O.; writing—review and editing, F.H. and M.O.; visualization, F.H. and M.O.; supervision, M.O.; project administration, F.H. and M.O.; funding acquisition, M.O. All authors have read and agreed to the published version of the manuscript.

**Funding:** This research was funded by the German Federal Ministry for Digital and Transport within the research project "ViVre" (Grant no.: 01MM19014A).

**Institutional Review Board Statement:** The study was conducted in accordance with the Declaration of Helsinki.

**Informed Consent Statement:** Informed consent was obtained from all subjects involved in the study.

**Acknowledgments:** We wish to thank Till Meyer-Arlt for supporting the design and development process of the vStop HMI prototype; Sarah Helweg, Felix Burgdorf, Till Meyer-Arlt and Hüseyin Avsar for supporting the phase of data collection; Till Meyer-Arlt and Hüseyin Avsar for supporting the concept phases of both user studies.

**Conflicts of Interest:** The authors declare no conflict of interest.

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
