# Peer review of "Design and Field Test of a Mobile Augmented Reality Human–Machine Interface for Virtual Stops in Shared Automated Mobility On-Demand"

_electronics, doi:10.3390/electronics11172687_

Round 1

Reviewer 1 Report

The paper presents a novel HMI interface for a mobility on-demand service for a smartphone using augmented reality. The HMI provides an intuitive navigation to the close by pick-up location with pathway elements (like breadcrumbs), indicates the virtual pick-up location vStop with an virtual area and an upright pole on the street environment and highlights the arriving automated vehicle with a symbol.

After the introductory presentation of the future mobility service with automated shared rides and the challenge of guiding people to the dynamic pick-up points and assigning them to the vehicles, it is described in detail how the development was accompanied by two studies. In the first online study users get introduced in the idea and possible designs were presented and evaluated. In the following a prototype was developed and a second study for the evaluation of this prototype was executed to assess user experience, acceptance and workload. Appropriate questionnaires and qualitative interview are used for evaluation. The implementation, the evaluation with the results and the interpretation of these are very detailed and accurate. The tables show a clear summary of the main results. In the discussion, the various limitations (small sample in the first study with young participants, evaluation with static pictures, reliability and stability of the prototype etc.) are well addressed and how these can be countered in the future.

The prototype was found to be very useful and helpful by the test persons, who were able to successfully complete the navigation task despite the time limit. A promising starting point for further research and improvements.

The paper clearly describes the problem and how the solution was developed. The methods and evaluations used are comprehensive and accurate. Detailed discussions of the results and findings are easy to follow and relevant. The final conclusions are consistent.

I noticed following errors:

Line 87: accompanyingying … one ing too much

Line 170: Rehrl et al., the year is missing

Line 614: Reference source not found

Line 738: Table 4…. The table caption should stand alone

Author Response

Dear Reviewer,

We wish to resubmit our manuscript entitled “Design and Field Test of a Mobile Augmented Reality Human-Machine Interface for Virtual Stops in Shared Automated Mobility On-demand” for publication in the journal Electronics Special Issue "Advances in Augmenting Human-Machine Interface". Thank you very much for the reviews and comments on our manuscript. The insights provided were truly valuable for perfecting our submission. We highly appreciate this!

Please find in the following our answers to the your comments (in italics). We have addressed them all to improve our manuscript. Thanks again, the reviewers’ comments were very valuable to us! We included the answers to each of the comments also in the revised version of the full paper manuscript (yellow highlighted passages and visible using the “Track Changes” mode of Microsoft Word).

English language and style are fine/minor spell check required

Answer: Thanks a lot. We conducted another check for English language.

The paper presents a novel HMI interface for a mobility on-demand service for a smartphone using augmented reality. The HMI provides an intuitive navigation to the close by pick-up location with pathway elements (like breadcrumbs), indicates the virtual pick-up location vStop with an virtual area and an upright pole on the street environment and highlights the arriving automated vehicle with a symbol.

Answer: Correct, thank you.

After the introductory presentation of the future mobility service with automated shared rides and the challenge of guiding people to the dynamic pick-up points and assigning them to the vehicles, it is described in detail how the development was accompanied by two studies. In the first online study users get introduced in the idea and possible designs were presented and evaluated. In the following a prototype was developed and a second study for the evaluation of this prototype was executed to assess user experience, acceptance and workload. Appropriate questionnaires and qualitative interview are used for evaluation. The implementation, the evaluation with the results and the interpretation of these are very detailed and accurate. The tables show a clear summary of the main results. In the discussion, the various limitations (small sample in the first study with young participants, evaluation with static pictures, reliability and stability of the prototype etc.) are well addressed and how these can be countered in the future.

Answer: Thank you very much.

The prototype was found to be very useful and helpful by the test persons, who were able to successfully complete the navigation task despite the time limit. A promising starting point for further research and improvements.

Answer: Thanks.

The paper clearly describes the problem and how the solution was developed. The methods and evaluations used are comprehensive and accurate. Detailed discussions of the results and findings are easy to follow and relevant. The final conclusions are consistent.

Answer: Thank you very much!

I noticed following errors:

  1. Line 87: accompanyingying … one ing too much.

Answer: Thank you very much for this note.

  1. Line 170: Rehrl et al., the year is missing.

Answer: We have added the year of publication here. Thanks a lot for this remark.

  1. Line 614: Reference source not found

Answer: Thank you, we have corrected the mistake (see line 616; in “track changes” mode).

  1. Line 738: Table 4…. The table caption should stand alone

Answer: Thank you very much for your remark. We edited the layout across the whole manuscript to fit all tables and table captions.

Thanks so much for your comments on our manuscript. We think that they were very helpful to improve the quality of the paper.

Reviewer 2 Report

A very well-written manuscript. It is not so often that I receive such quality written manuscripts for review. Great job!

Maybe, the manuscript could be divided into two separate articles, since it is a little bit longer and reports the results of two studies. However, since the studies are connected, it is better for a reader to get acquainted with the results of both while reading the same article.

The backgrounds, the methodology, procedures, and results are very thoroughly presented and the results are properly discussed together with the study's limitations.

In the text, there are some minor grammar mistakes, which will be resolved with the help of the journal's proofreading service. For example, in Line 614, there is a cross-reference error, which I suppose the authors already noticed. 

In the limitations section, the authors should also add a discussion of limitations regarding the studies' results, sample, etc. Furthermore, threats for internal and external reliability and validity could also be discussed, and what measures were taken to minimize different threats.

Although the authors stated that for getting users' feedback standard questionnaires were used, I would prefer if the questionnaires used for collecting empirical data in these two studies were added either in the appendices section or as supplementary material.

Author Response

Dear Reviewer,

We wish to resubmit our manuscript entitled “Design and Field Test of a Mobile Augmented Reality Human-Machine Interface for Virtual Stops in Shared Automated Mobility On-demand” for publication in the journal Electronics Special Issue "Advances in Augmenting Human-Machine Interface". Thank you very much for the reviews and comments on our manuscript. The insights provided were truly valuable for perfecting our submission. We highly appreciate this!

Please find in the following our answers to your comments (in italics) in this letter. We have addressed them all to improve our manuscript. Thanks again, the reviewers’ comments were very valuable to us! We included the answers to each of the comments also in the revised version of the full paper manuscript (yellow highlighted passages and visible using the “Track Changes” mode of Microsoft Word).

Moderate English changes required

Answer: Thanks a lot. We conducted another check for English language.

  1. A very well-written manuscript. It is not so often that I receive such quality written manuscripts for review. Great job!

Answer: Thank you very much. We really appreciate your kind words.

  1. Maybe, the manuscript could be divided into two separate articles, since it is a little bit longer and reports the results of two studies. However, since the studies are connected, it is better for a reader to get acquainted with the results of both while reading the same article.

Answer: Thanks for articulating your thought here. We had the same considerations but eventually decided to put both studies in one joint article. Since both studies build on each other, we believe that the reader should be able to follow the development process thoroughly.

  1. The backgrounds, the methodology, procedures, and results are very thoroughly presented and the results are properly discussed together with the study's limitations.

Answer: Thank you very much.

  1. In the text, there are some minor grammar mistakes, which will be resolved with the help of the journal's proofreading service. For example, in Line 614, there is a cross-reference error, which I suppose the authors already noticed.

Answer: Thanks a lot for your remark. We conducted another check for English language and corrected the cross-reference error (see line 616; in “track changes” mode).

  1. In the limitations section, the authors should also add a discussion of limitations regarding the studies' results, sample, etc. Furthermore, threats for internal and external reliability and validity could also be discussed, and what measures were taken to minimize different threats.

Answer: Thank you very much for this remark. We added aspects regarding internal and external reliability and validity to the limitations section (see lines 975-994; in “track changes” mode).

  1. It is not clear what to do with this qualitative information in study 1. Also, many pain points would be outside of the service’s control (cancellation of the trip or changes in pick-up times).

Answer: Thank you very much for this note. We are not completely sure how to deal with this comment and how it is linked to the first user study. In the second user study however, capturing qualitative information was included to provide a broader view of the exploratory insights from users’ perspectives to the reader. Nevertheless, we totally agree with you that specific user pain points like trip cancellation or delays are somewhat outside of the service providers control. How this information should be presented to users may be subject of further HMI research. Eventually, these aspects could also have implications on SAMOD acceptance.

  1. Although the authors stated that for getting users' feedback standard questionnaires were used, I would prefer if the questionnaires used for collecting empirical data in these two studies were added either in the appendices section or as supplementary material.

Answer: Thanks for highlighting this point. Due to the length of the article and the fact that all used questionnaires are published and well cited, we would like to keep it this way (and provide references). If we would have used own items /questionnaires we would have added these in the appendices for sure.

Thanks so much for your comments on our manuscript. We think that they were very helpful to improve the quality of the paper.